# Modeling and Control of an Underactuated System for Dynamic Body Weight Support

**Grzegorz Gembalczyk** [1,*] , **Piotr Gierlak** [2] **and Slawomir Duda** [1]

1 Department of Theoretical and Applied Mechanics, Faculty of Mechanical Engineering, Silesian University of Technology, Akademicka 2A, 44-100 Gliwice, Poland; Slawomir.Duda@polsl.pl

2 Department of Applied Mechanics and Robotics, Faculty of Mechanical Engineering and Aeronautics, Rzeszow University of Technology, 35-959 Rzeszów, Poland; pgierlak@prz.edu.pl

\* Correspondence: Grzegorz.Gembalczyk@polsl.pl; Tel.: +48-32-237-24-62

**Featured Application: The results of the work may find application in the stability analysis of devices for active body weight support during gait re-education.**

**Abstract:** This article concerns the stability analysis of a control system for a dynamic body weight support system in a rehabilitation device for the re-education of human gait. The paper presents a physical model of the device, which characterizes the most important physical phenomena associated with the movement of the system, i.e., inertia, damping, and elasticity. The device has one active and one passive element. They are connected by a connector with elastic and damping properties. This solution provides the kinematic chain required due to interactions with humans, while at the same time ensures that the device is an underactuated system. The article also presents the methodology used to verify the stability of the control system while acting as an active body weight support system. The paper formulates the mathematical model of the system that was used in the synthesis of control using the Lyapunov theory of stability. The results of simulation and experimental tests are also presented.

**Keywords:** body weight support; rehabilitation engineering; robot-assisted gait training; dynamics modeling; control systems; stability analysis; Lyapunov theory

## 1. Introduction

The analysis of devices for rehabilitation purposes related to kinesitherapy designed during the last dozen years has indicated that body weight support systems are a crucial component. At present, there are many simple devices available on the market that are passive systems, as well as technologically advanced equipment fitted with an active, mechatronic body weight support system. Active systems enable a defined support load to be maintained, giving high accuracy, whereas passive systems (load value manually adjusted with a lever, counterweight, spring, etc.) often cause overload and a large difference between the adjusted load setting and recorded load setting during training. Therefore, devices fitted with an active body load support system are more favorable for the proper performance of exercises [1–3]. During the re-education of human gait, these systems are used to reduce body weight and reduce leg loading at the same time. A person performing exercises is clamped to the rope end of a winch with an orthopedic harness.

When comparing rehabilitation devices fitted with an active load support system, the stationary element is mostly related to the fixed load bearing structure (it does not follow patient movements) and human gait is forced by mechanical orthosis or a treadmill [4–11]. In devices with a treadmill, training is recommended for persons who have restored their gait to a degree, enabling unaided movement, but who still require assistance, or when gait manner, pace, and strength need improvement [12,13]. Such devices provide technological advancements related to virtual reality and biofeedback [14–18].

One of the latest kinesitherapy devices is a mechatronic treadmill with a body weight support system developed by the Silesian University of Technology [19]. The active body weight support system is similar to other technologically advanced winches in rehabilitation devices for the re-education of human gait [1,20–22]. It has two independent drives: a linear drum and series elastic actuator (SEA)-type drive. Because the range of vertical human body movements is rather small during gait, this system is only a body weight dynamic compensation system. Specifically, the SEA drive maintains the load according to the programmed settings, whereas the rotating drum operation is used when big vertical movements occur, e.g., upright standing, knee bends, walking up stairs.

The SEA system features a specific design including the kinematic chain adaptability required due to interactions with humans, and it protects against overloading resulting from a delayed servodrive response [23–25]. The SEA system includes one active part (controlled) and one passive part, which are connected by a connector with elastic and damping properties. Thanks to this solution, the device is an underactuated system, whereby the number of generalized variables is greater than the number of programmable variables [26–29].

Note that the SEA drive minimizes offset-related support load errors in a reliable and quick manner to prevent overload during the performance of exercises. Moreover, any hazards to human health are prevented. The development of a control system that meets both requirements poses an interesting engineering problem. Considering the necessity to ensure the reliable operation of the device during any situation, control systems used for that purpose offer stability.

An analysis of current papers on similar devices for the re-education of gait indicated that these systems most often employ a classic feedback loop and the settings of controllers are programmed by way of experiments. No previous paper focused on studies regarding stability. Therefore, the current paper is innovative as it looks at the development of a control algorithm for a body weight support system that takes into account stability analysis according to the Lyapunov theory of stability.

Determining control system stability is a very difficult task that sometimes requires a non-standard mathematical description of the tested object to be formulated. Relevant experience in this field is invaluable. This paper presents the methodology used to verify a control system that can act as a dynamic body weight support system, where instead of the typical task of maintaining a constant body support force value, it maintains a set trajectory. The methodology applied for the analysis of stability provided a motive for verifying a new approach (not yet described in the literature) regarding the control system of a rehabilitation device for the re-education of human gait, where patient gait kinematics are forced by the body weight support system. The novelty of the presented control system is related to the possibility of using the function describing a patient's gait pattern. This approach is particularly interesting due to the results of previous studies, which despite their high effectiveness also suggest that an incorrect gait pattern can be consolidated while training on a treadmill under unloaded conditions when the active body weight support system works only in the constant force mode [30–33].

Contrary to the control algorithms used previously, the novelty of this article is the use of the proportional-derivative (PD) controller with compensation related to the dynamic properties of the passive part in the body weight support system. The use of PD regulators with dynamic compensation is a well-known issue, but their use in devices intended to support gait re-education has not been presented in the literature so far. This publication shows the successive stages involved in designing such a control algorithm, starting from the formulation of the object equations of motion and moving on to the definition of the control law and the determination of the set motion trajectory of the active part of the system. Due to the use of an underactuated SEA drive, the proposed control system for the dynamic body weight support system performs better than the standard feedback loop provided with the PD controller.

Chapter 2 describes the test bench, considering in particular the sensors and method used for the processing of measurement signals in real time. Next, a mathematical model for the system is presented and applied for the analysis of the control system on the basis of the Lyapunov theory of stability. Then, certain mathematical calculations that follow the design phase of the control system and stability tests are presented. The next part of the paper includes the results of simulation tests, empirical studies, and conclusions.

## 2. Methodology and Research Object

The purpose of this paper is to present the methodology used for the stability analysis of a control system for a dynamic body weight support system. A PD controller with compensation was proposed to control the drive system. As an example of a device for the re-education of gait, a body weight support system coupled with a treadmill was used [19]. The main components of the electromechanical part of the rehabilitation system are shown in Figure 1.

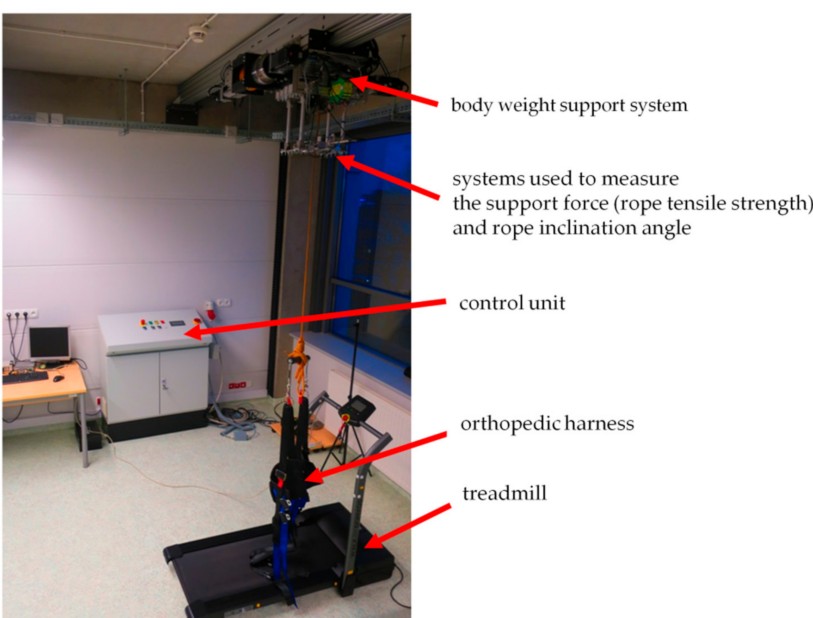

**Figure 1.** Mechatronic device for gait re-education.

The structure of the body support system was analyzed in detail in [34]. Although the body support system is fitted with two independent drive systems, the presented analyses assumed that the dynamic compensation system operated only when gait was measured on a treadmill due to the little vertical movement of the patient during gait, with values falling within the range of SEA drive parameters.

For maintenance of a constant body support load, one sensor that controls the drive operation with a feedback loop according to a signal indicating load offset is sufficient. The method proposed for the verification of the control system stability requires all generalized coordinates to be measured. For this purpose, additional adapters were developed for spring deflection measurement of the SEA system and determination of the drive's passive component position. Measurements were performed with a linear variable differential transformer sensor. Figure 2 presents the crucial components of the dynamic compensation system.

Control was measured in real time using MATLAB/Simulink branded software. Transmission of the control and measurement signals was done with two RT-DAC4 cards provided by the Spartan II series Field-Programmable Gate Array (FPGA) system. Communication with the system was done with system Peripheral Component Interconnect (PCI) bus and external input/output accelerator (PLX 9030). FPGA along with programming languages enabled us to transfer operations related to complicated digital processing of

measurement data and control signal modulation onto hardware. An interface for analogue signal conditioning was an additional part of the measuring system. Drive control signals generated by the control application were sent to particular servodrives in the form of voltage at an appropriate level. The servomotors were controlled in speed mode. The signal sent from the computer was proportional to the set rotational speed and the servoinverter controlled the motor torque [35].

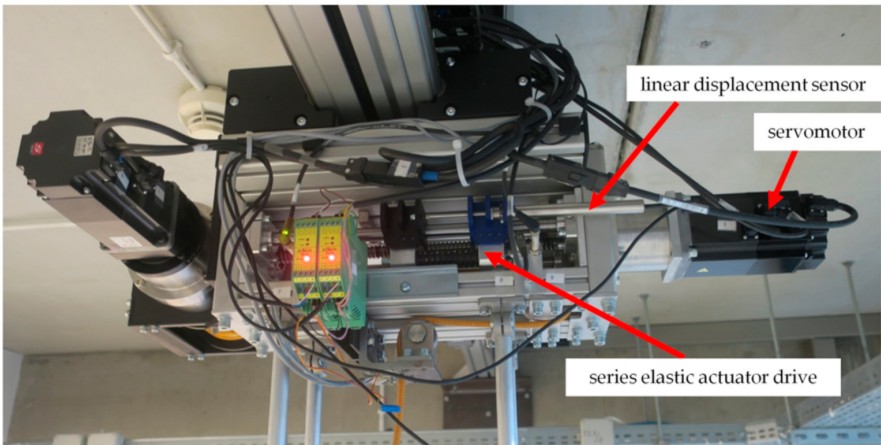

**Figure 2.** Body weight support system.

To demonstrate the stability of the control system, it was necessary to formulate a set of equations regarding the movement of mechanical parts, develop a control algorithm, determine relevant errors, and define the control method. Next, determination of the Lapunov function was necessary. These steps are described in the following chapters of this paper.

### 3. Dynamic Equations of Motion

The formulation of equations regarding motion development was necessary to construct the physical model of the device. The performed calculations were based on an essential simplifying assumption that considered the rope to be a stiff and non-tensile component. This is due to the complexity of the rope modeling process (especially using synthetic ropes as multi-fiber bodies) and the fact that the ropes were characterized by very high stiffness with increasing tension and number of load cycles [36–40]. The adopted assumption is justified only in the case of a control system that performs well and ensures that constant load support related directly to rope tension and deformation is maintained. If the control system does not perform the desired motion with a high level of accuracy, this assumption cannot be made and the physical model must also include the rope. Figure 3 presents the physical model adopted for the body weight support system.

The dynamic performance of the adopted physical model can be described by differential equations of motion. The mathematical model was developed using Lagrange's equations of the second kind, which can be written as:

$$\begin{cases} I_Z \ddot{\varphi} + \left( b_Z + b_s \frac{h^2}{4\pi^2} \right) \dot{\varphi} - b_s \frac{h}{2\pi} \dot{x} + k_s \frac{h^2}{4\pi^2} \varphi - k_s \frac{h}{2\pi} x = M_Z \\ m_Z \ddot{x} + b_s \dot{x} - b_s \frac{h}{2\pi} \dot{\varphi} + k_s x - k_s \frac{h}{2\pi} \varphi = 2F \end{cases} \tag{1}$$

where $\varphi$ is the drive motor shaft rotation angle, $x$ is the displacement of the passive part, $I_Z$ is the reduced moment of inertia of the active part, $m_Z$ is the reduced mass of the passive part, $b_Z$ is the simplified ratio of the active part's motion resistance, $b_s$ is the damping system ratio, $k_s$ is the spring stiffness modulus, $h$ is the thread size of the screw drive system, $M_Z$ is the servomotor torque, and $F$ is the set load of the body weight support.

The system is determined with variables $\varphi$ and $x$, whereas the rotation angle $\varphi$ is the programmable variable.

The mathematical model for the relieving system with 3 degrees of freedom is described in [34].

A problematic issue is related to the impact of patient body inertia. During training, the patient's body is supported with a vertical load (rope tension) that provides weight reduction. Considering the patient's inertia related to their whole body mass is an incorrect approach, as such a system cannot be considered synonymously to the patient's whole-body support. Ignoring their mass is also not justified, as mass has a significant impact on the device's dynamic performance. Therefore, calculations were performed to determine whether the patient's mass affected the reduced mass of the passive part proportionally to the adjusted body support force.

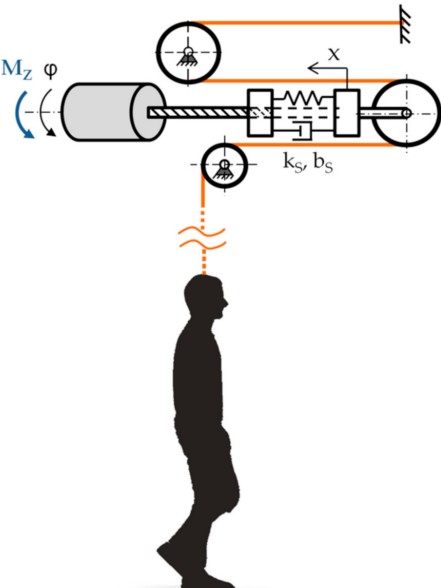

**Figure 3.** Physical model of the dynamic body weight support system.

## 4. Control System Modeling

The actual purpose of the control system of a support load system is to maintain constant load support. Constant rope tension relates to constant deflection of the rope. Therefore, an assumption may be made that the issue related to keeping a constant support load is synonymous to follow-up motion related to vertical movement of the patient. According to the presented model, these movements directly impact the movement of the passive part, namely generalized coordinate $x$.

Further calculations were based on an adjusted trajectory of the passive part $x_d(t)$ featuring continuous derivatives, and an assumption was made regarding the existence of the adjusted trajectory of the active part $\varphi_d(t)$ with continuous derivatives [41]. Next, the trajectory error was determined:

$$
\begin{cases}
e_a = \varphi_d - \varphi \\
e_p = x_d - x
\end{cases}
\tag{2}
$$

where $e_a$ is the follow-up movement error of the active part and $e_p$ is the follow-up movement error of the passive part. The generalized follow-up movement error was determined using:

$$
\begin{cases}
s_a = \dot{e}_a + \lambda_a e_a \\
s_p = \dot{e}_p + \lambda_p e_p
\end{cases}
\tag{3}
$$

where $\lambda_a$ and $\lambda_p$ indicate positive design constants related to the ratio of proportional and differential gains in the PD controller.

Equation (3) considers Equation (2). Through conversion and derivation, the acceleration was determined:

$$\begin{cases} \ddot{\varphi} = -\dot{s}_a + \ddot{\varphi}_d + \lambda_a \dot{e}_a = -\dot{s}_a + \dot{v}_a \\ \ddot{x} = -\dot{s}_p + \ddot{x}_d + \lambda_p \dot{e}_p = -\dot{s}_p + \dot{v}_p \end{cases} \tag{4}$$

where $v_a = \dot{\varphi}_d + \lambda_a e_a$ and $v_p = \dot{x}_d + \lambda_p e_p$ are additional variables applied to simplify recording.

The application of (4) to (1) resulted in a description of the generalized follow-up movement error function:

$$\begin{cases} I_Z \dot{s}_a = I_Z \dot{v}_a + \left( b_Z + b_s \frac{h^2}{4\pi^2} \right) \dot{\varphi} - b_s \frac{h}{2\pi} \dot{x} + k_s \frac{h^2}{4\pi^2} \varphi - k_s \frac{h}{2\pi} x - M_Z \\ m_Z \dot{s}_p = m_Z \dot{v}_p + b_s \dot{x} - b_s \frac{h}{2\pi} \dot{\varphi} + k_s x - k_s \frac{h}{2\pi} \varphi - 2F \end{cases} \tag{5}$$

Expressions related to the system's condition were determined with the following functions:

$$\begin{cases} f_a = I_Z \dot{v}_a + \left( b_Z + b_s \frac{h^2}{4\pi^2} \right) \dot{\varphi} - b_s \frac{h}{2\pi} \dot{x} + k_s \frac{h^2}{4\pi^2} \varphi - k_s \frac{h}{2\pi} x \\ f_p = m_Z \dot{v}_p + b_s \dot{x} - b_s \frac{h}{2\pi} \dot{\varphi} + k_s x - k_s \frac{h}{2\pi} \varphi - 2F \end{cases} \tag{6}$$

Finally, the following set of Equation (5) was determined:

$$\begin{cases} I_Z \dot{s}_a = f_a - M_Z \\ m_Z \dot{s}_p = f_p \end{cases} . \tag{7}$$

To obtain the adjusted trajectory of motion, an assumption was made regarding the control inputs of the proportional-derivative controller with compensation:

$$\begin{cases} u_a = k_a s_a + f_a \\ u_p = k_p s_p + f_p \end{cases} \tag{8}$$

where $u_a$ is the control of the active part equal to $u_a = M_Z$ and $u_p$ is the fictional control of the passive part equal to $u_p = 0$ [42]. Here, $k_a s_a = k_a (\dot{e}_a + \lambda_a e_a)$ and $k_p s_p = k_p (\dot{e}_p + \lambda_p e_p)$ describes the PD controller for the active and passive parts, respectively, where $k_a$ and $k_p$ improve the derivation.

The assumed control method in (8) includes two components, whereas the second equals zero as the passive part is not controlled. This component is called fictional control, as it does not have a direct impact on the movement of the second part of the system. However, it determines the dynamic performance of the passive part as it contains function $f_p$. Therefore, it can be used to determine the adjusted trajectory of the active part $\varphi_d(t)$, which will ensure the desired performance of the passive part is achieved, namely trajectory $x_d(t)$.

The equation used to determine fictional control considering the $f_p$ function determined with Equation (6) is as follows:

$$k_p s_p + m_Z \dot{v}_p + b_s \dot{x} - b_s \frac{h}{2\pi} \dot{\varphi} + k_s x - k_s \frac{h}{2\pi} \varphi - 2F = 0. \tag{9}$$

This provides the basis to determine the speed of the active part:

$$\dot{\varphi} = \frac{1}{b_s \frac{h}{2\pi}} \left( k_p s_p + m_Z \dot{v}_p + b_s \dot{x} + k_s x - k_s \frac{h}{2\pi} \varphi - 2F \right) \tag{10}$$

with velocity applied in (10) as a set speed of motion:

$$\dot{\varphi}_d = \dot{\varphi} \tag{11}$$

The integration of Equation (11) results in determination of the set trajectory of the active part:

$$\varphi_d = \int_0^t \dot{\varphi}_d dt. \tag{12}$$

The control system ensures that the trajectory of the active part is adjusted, namely that it controls the motor shaft rotation angle.

## 5. Closed System Stability

The stability of the closed system was determined on the basis of the Lyapunov theory of stability. For this purpose, a square form of the generalized follow-up motion error was applied using the Lapunow approach:

$$L = \frac{1}{2} I_Z s_a^2 + \frac{1}{2} m_Z s_p^2. \tag{13}$$

Because simplified inertia and simplified mass relations are constants, a derivative of Equation (13) in relation to time is equal to:

$$\dot{L} = I_Z s_a \dot{s}_a + m_Z s_p \dot{s}_p. \tag{14}$$

Considering the description of the controlled object, namely Equation (7), a derivative of the $L$ function for determining the system's trajectory was determined:

$$\dot{L} = s_a(f_a - M_Z) + s_p f_p. \tag{15}$$

Considering the control law $M_Z = u_a = k_a s_a + f_a$, Equation (15) was formulated as the following:

$$\dot{L} = -k_a s_a^2 + s_p f_p. \tag{16}$$

Function $f_p$, determined with (6) and taking into account (10), is expressed as:

$$f_p = -k_p s_p. \tag{17}$$

This also results from the equation determining the fictional control $u_p = k_p s_p + f_p = 0$. Applying the above in Equation (16) resulted in:

$$\dot{L} = -k_a s_a^2 - k_p s_p^2 \leq 0. \tag{18}$$

Because function $L$ is positive with reference to the generalized follow-up movement error and the conducted analysis indicated that derivative $\dot{L}$ is a negative semidefinite, variables $s_a$ and $s_p$ are limited pursuant to the Lyapunov theory of stability. On the basis of Barbalat's lemma, you can determine the convergence to zero of $s_a$ and $s_p$ as the result of the convergence to zero of errors $e_a$ and $e_p$.

## 6. Simulation Research

In terms of numerical research, some simulation tests were performed regarding the motion of the rehabilitation device on the basis of Equations (1) and (8). Tables 1 and 2 present the data applied during the simulation tests.

**Table 1.** Model parameters applied during simulation and experimental tests.

| Parameter | Unit | Value |
|-----------|------|-------|
| $I_Z$ | kg m$^2$ | 0.00018 |
| $m_Z$ | kg | 10.5 + 2F/g |
| $b_Z$ | kg m$^2$/s | 0.0005 |
| $b_s$ | kg/s | 300 |
| $k_s$ | N/m | 96,800 |
| $h$ | mm | 0.005 |

**Table 2.** Control system parameters applied during simulation and experimental tests.

| Parameter | Unit | Value |
|-----------|------|-------|
| $\lambda_a$ | 1/s | 120 |
| $\lambda_p$ | 1/s | 6200 |
| $k_a$ | kg m$^2$/s | 3 |
| $k_p$ | kg/s | 120 |

The trajectory of the passive part was specified on the basis of two variants. In the first case, simulation tests were based on an analysis regarding the kinematics of the gait patterns of persons diagnosed with neurological diseases. The averaged motion along the vertical axis for one tested person was determined with (19). Therefore, this motion may be considered to be a gait template for a disabled person at a given phase of rehabilitation. This motion relates to the rope end displacement also. Pursuant to the device structure, a value of the generalized variable $x$, which determines the chair movement within the passive part of the body weight support system, is reduced twice compared with the patient's vertical movement. Consequently, Function (20) determines the adjusted trajectory of the body weight support system's passive part.

$$F_p(t) = 0.016(\sin(2\pi t) - \sin(4\pi t)) \tag{19}$$

$$x_d(t) = 0.008(\sin(2\pi t) - \sin(4\pi t)). \tag{20}$$

In the second case, based on literature data [43,44], the function that determines the adjusted trajectory of the body weight support system's passive part related to vertical movement during a healthy person's gait (gait template) was defined. This function was determined with Equation (21):

$$x_d(t) = 0.006(\sin(4\pi t)). \tag{21}$$

The trajectory of the body weight support system's active part is generated on an ongoing basis during device operation.

The performed simulation tests considered three cases:

- Variant 1: A person during rehabilitation performs movements pursuant to (19) without body weight support;
- Variant 2: During rehabilitation, a person performs movements pursuant to (19) with body weight support of 300 N;
- Variant 3: During rehabilitation, a person performs movements with body weight support of 300 N.

The simulation tests assumed that the body weight of the person undergoing rehabilitation was 70 kg (the device has a load capacity of 150 kg).

### 6.1. Variant 1

Figure 4 shows the adjusted and recorded movements of the active and passive parts, while Figure 5 presents the adjusted and recorded movement speeds. Figure 6 presents the

total control signal along with the PD controller signal and compensation control signal. Figure 7 specifies the follow-up movement errors.

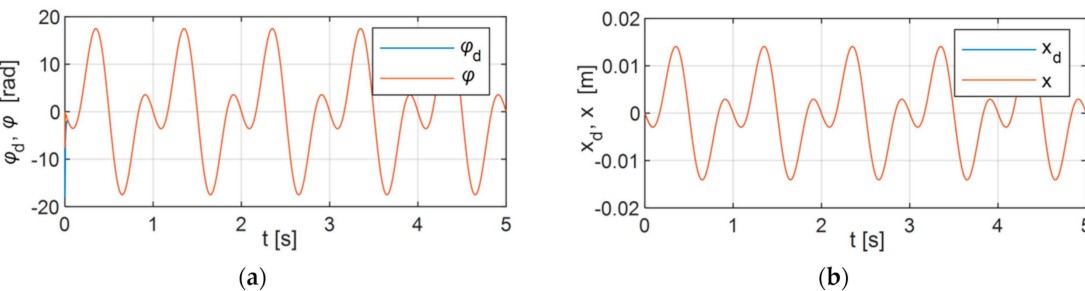

**(a)**　　　　　　　　　　　　　　　　　　**(b)**

**Figure 4.** Trajectory: (**a**) adjusted $\varphi_d$ and realized $\varphi$ positions of the active part; (**b**) adjusted $x_d$ and realized $x$ positions of the passive part.

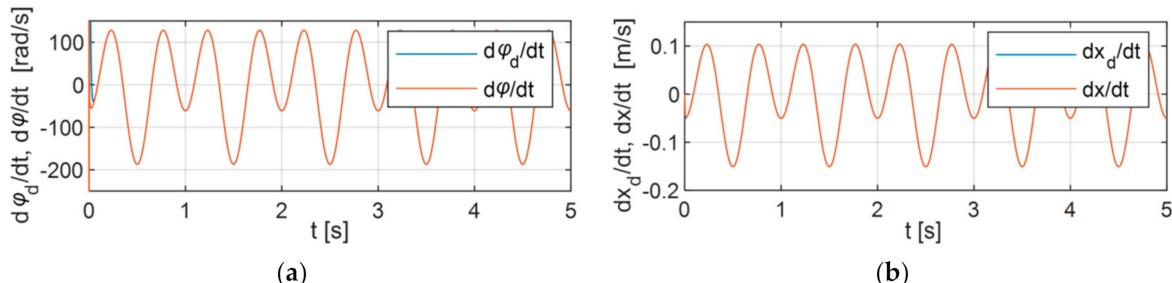

**(a)**　　　　　　　　　　　　　　　　　　**(b)**

**Figure 5.** Trajectory: (**a**) adjusted $d\varphi_d/dt$ and realized $d\varphi/dt$ speeds of the active part; (**b**) adjusted $dx_d/dt$ and realized $dx/dt$ speeds of the passive part.

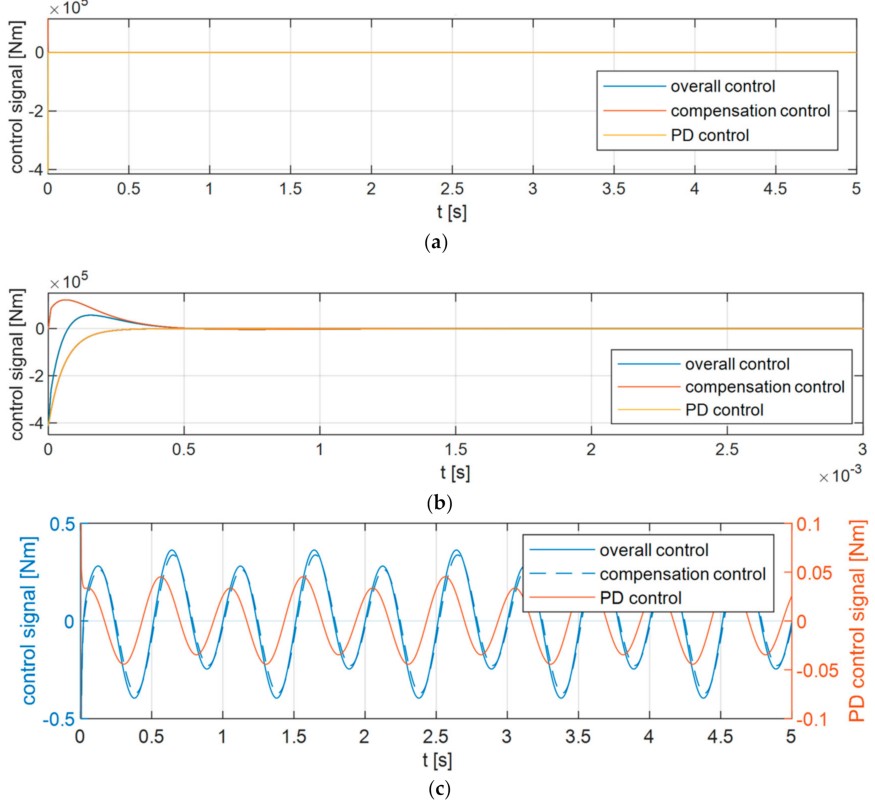

**(a)**

**(b)**

**(c)**

**Figure 6.** Control signals: (**a**) overall control system and control signal components; (**b**) control during the initial phase of gait; (**c**) control graph with customized scales.

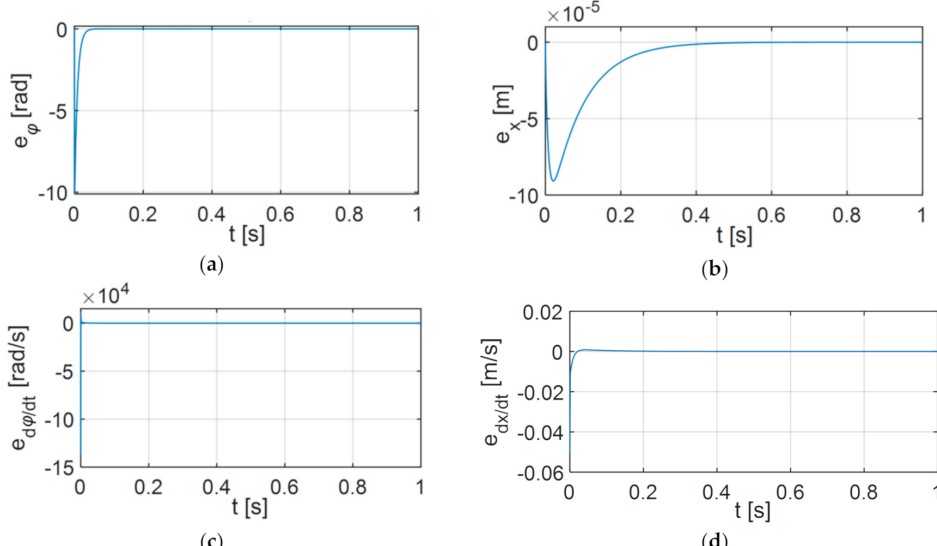

**Figure 7.** Follow-up movement errors: (**a**) position error of the active part $e_\varphi$; (**b**) position error of the passive part $e_x$; (**c**) speed error of the active part $e_{d\varphi/dt}$; (**d**) speed error of the passive part $e_{dx/dt}$.

The presented results relate to a situation where a person undergoing re-education maintains a vertical position on an unaided basis and performs the template gait during a given phase of rehabilitation. In this way, no disruption follows. In this situation, the device is not exposed to human body weight and the patient's mass does not affect the system's inertia. This example relates to a human aiming to improve their gait pattern. The adjusted trajectory errors result from the system's dynamic performance only. This case, however, needs to undergo verification with regard to the control system's operation.

### 6.2. Variant 2

The results of the second simulation test relate to the use of a rehabilitation device with a body weight support system that stimulates movement in the vertical direction. In this case, the patient's body mass affects the inertia of the system, which is exposed to additional loading caused by the human body weight. During kinesitherapy treatment, a body weight support load that reduces the effective load to 50% of the patient's body mass is applied [45–48]. Higher loads are not used in practice. The described simulation tests were performed with a load of 300 N. Therefore, an assumption was made that the system's inertia affected by the patient's body mass was 30 kg. Figure 8 shows the adjusted and recorded movements of the active and passive parts, while Figure 9 presents the adjusted and recorded movement speeds. Figure 10 presents the total control signal along with the PD controller signal and compensation control signal and Figure 11 specifies the follow-up movement errors.

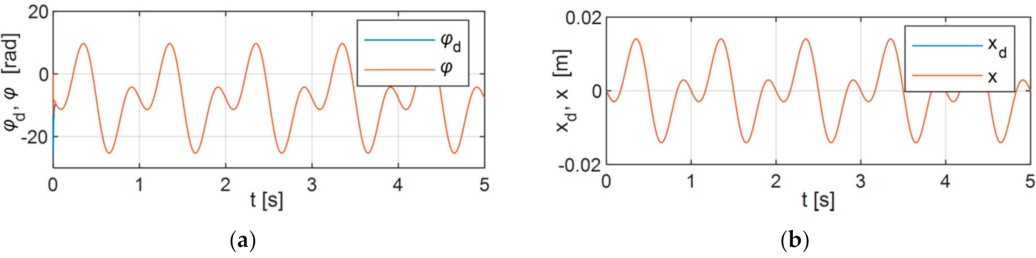

**Figure 8.** Trajectory: (**a**) adjusted $\varphi_d$ and realized $\varphi$ positions of the active part; (**b**) adjusted $x_d$ and realized $x$ positions of the passive part.

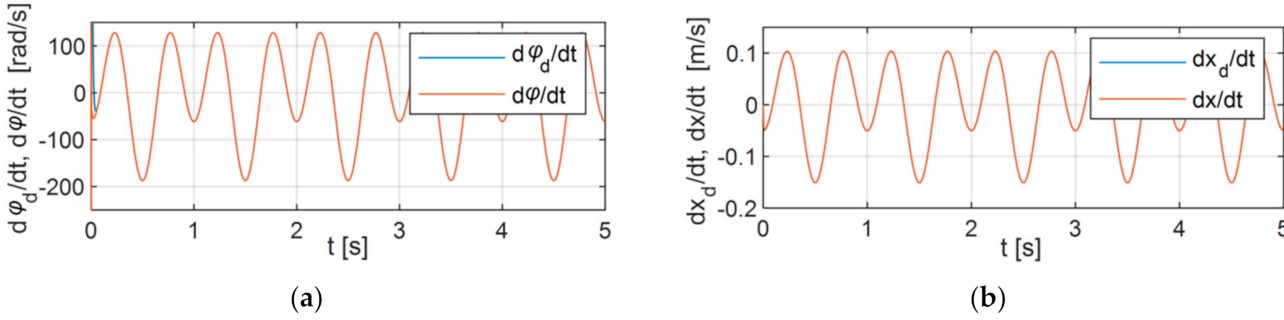

**Figure 9.** Trajectory: (**a**) adjusted $d\varphi_d/dt$ and realized $d\varphi/dt$ speeds of the active part; (**b**) adjusted $dx_d/dt$ and realized $dx/dt$ speeds of the passive part.

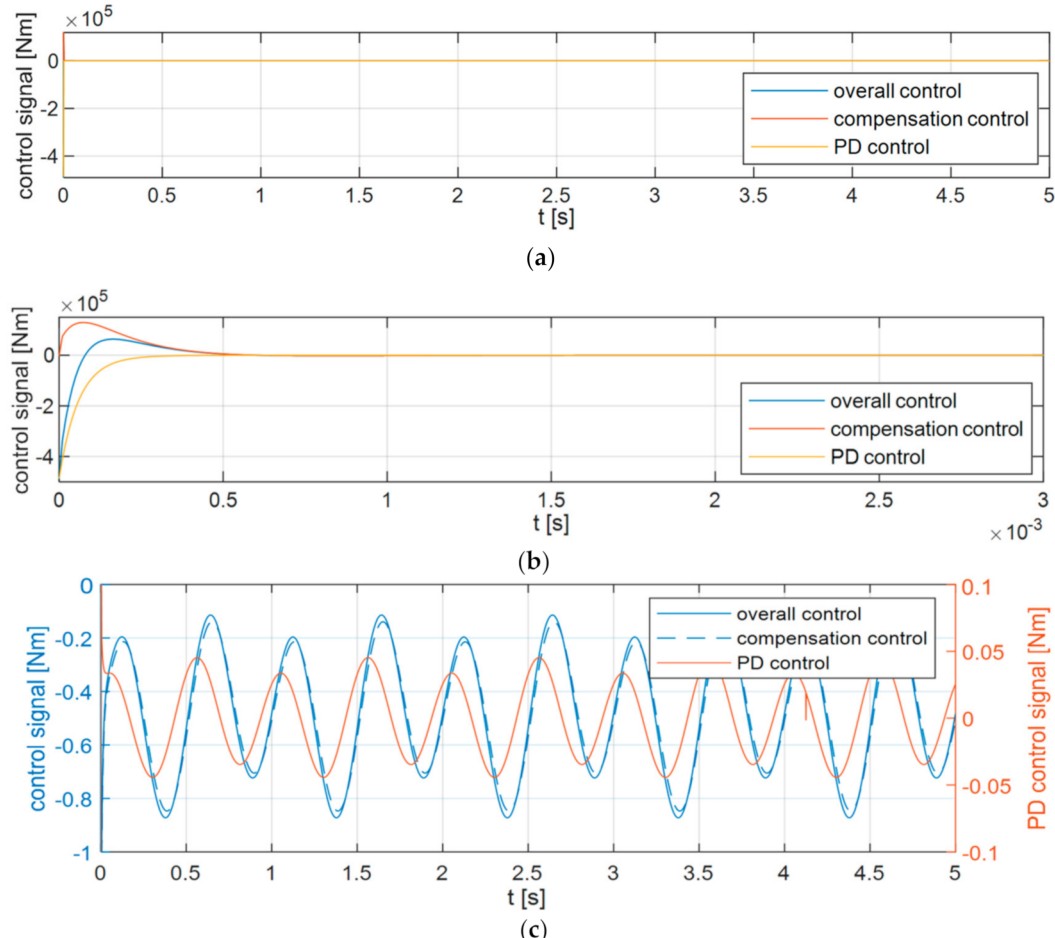

**Figure 10.** Control signals: (**a**) overall control system and control signal components; (**b**) control during the initial phase of gait; (**c**) control graph with customized scales.

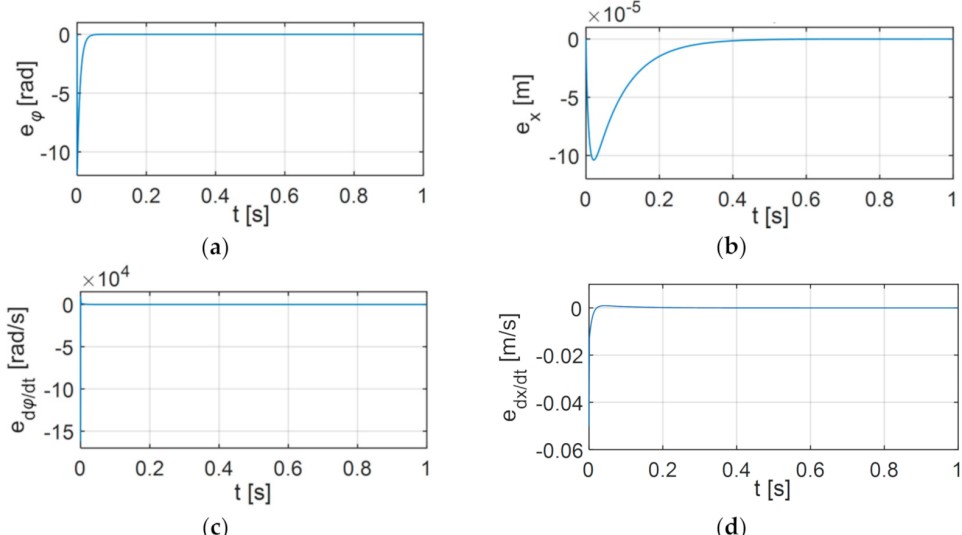

**Figure 11.** Follow-up movement errors: (**a**) position error of the active part $e_\varphi$; (**b**) position error of the passive part $e_x$; (**c**) speed error of the active par $e_{d\varphi/dt}$; (**d**) speed error of the passive part $e_{dx/dt}$.

### 6.3. Variant 3

The third analyzed case related to a situation analogous to variant 2, whereby a person undergoing rehabilitation tries to move and follow their natural gait. For this variant Figure 12 shows the adjusted and recorded movements of the active and passive parts, while Figure 13 presents the adjusted and recorded movement speeds. Figure 14 presents the total control signal along with the PD controller signal and compensation control signal. Figure 15 specifies the follow-up movement errors.

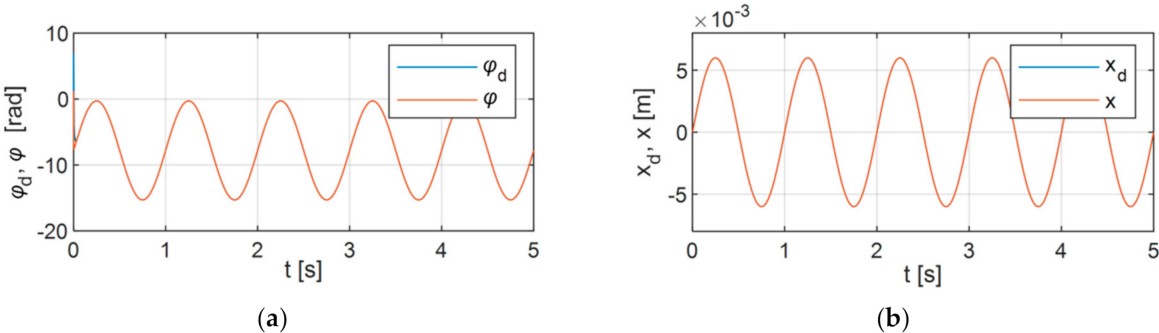

**Figure 12.** Trajectory: (**a**) adjusted $\varphi_d$ and realized $\varphi$ positions of the active part; (**b**) adjusted $x_d$ and realized $x$ positions of the passive part.

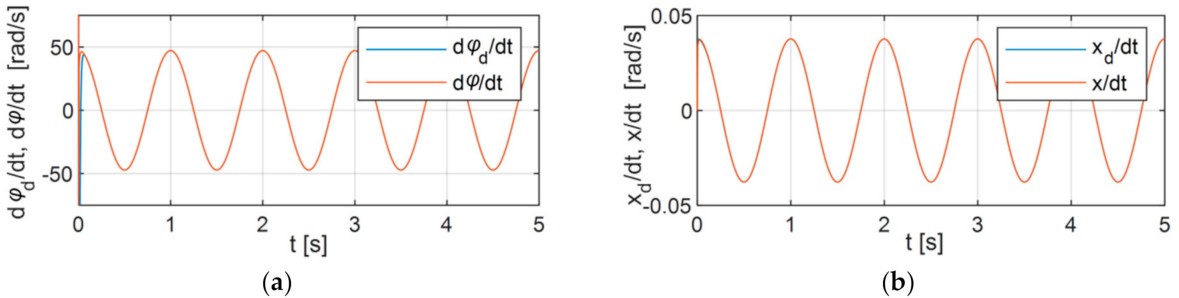

**Figure 13.** Trajectory: (**a**) adjusted $d\varphi_d/dt$ and realized $d\varphi/dt$ speeds of the active part; (**b**) adjusted $dx_d/dt$ and realized $dx/dt$ speeds of the passive part.

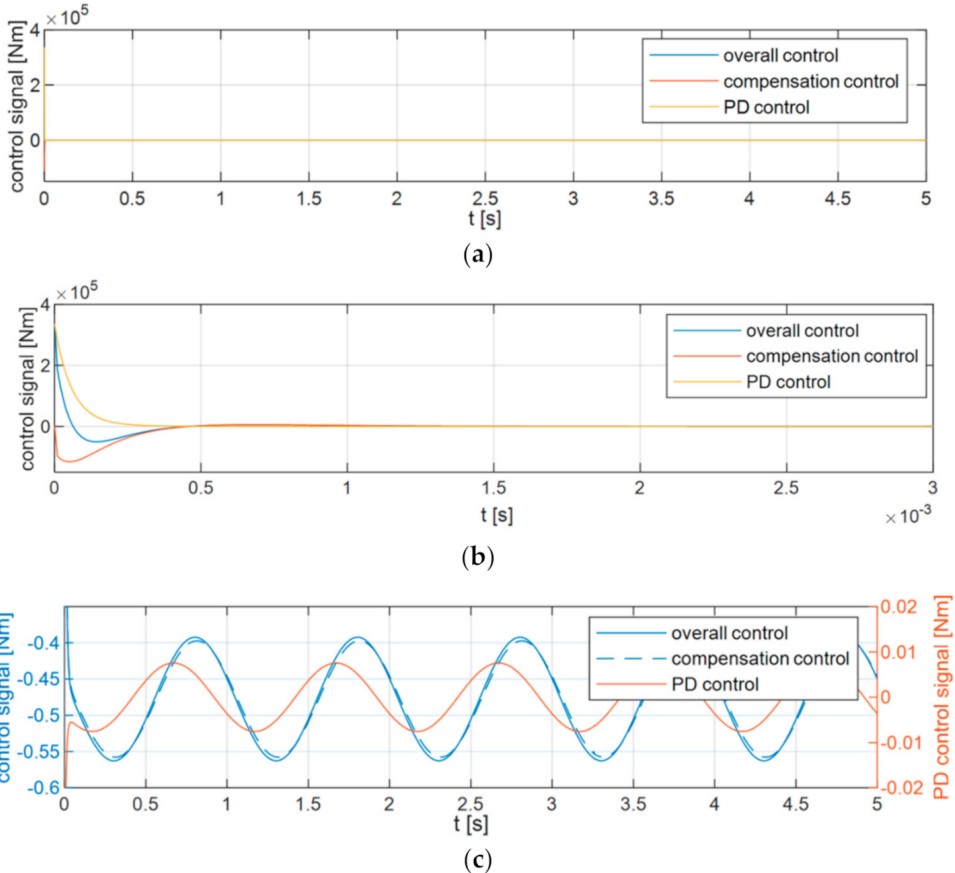

**Figure 14.** Control signals: (**a**) overall control system and control signal components; (**b**) control during the initial phase of gait; (**c**) control graph with customized scales.

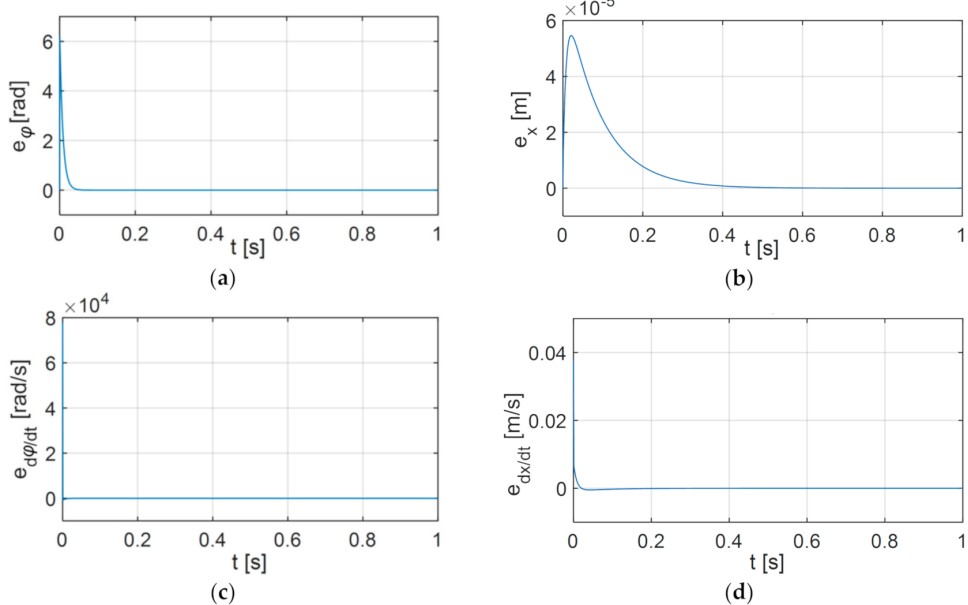

**Figure 15.** Follow-up movement errors: (**a**) position error of the active part $e_\varphi$; (**b**) position error of the passive part $e_x$; (**c**) speed error of the active part $e_{d\varphi/dt}$; (**d**) speed error of the passive part $e_{dx/dt}$.

### 6.4. Analysis of the Results

The results of simulation tests indicate that the designed control system meets our objectives and ensures stable operation, regardless of patient movement and the adjusted

weight support load. For the demonstrated approach, the trajectory of the passive part is adjusted over time and the trajectory of the active part can be calculated on the basis of (10)–(12). Control method (8) enables the adjusted trajectory of the passive part to be determined, because the dynamic performance of this system in the adjusted trajectory of the active part is considered. Comparing the results of the simulation tests for the first two variants, we observed differences mainly in the calculated set trajectory of the active part. For the variant with a support force of 300 N, the position of the active part corresponded to the deformation of the spring in the body weight support system, which is needed to compensate for this support force. We also identified differences in the values of the control signal corresponding to the servomotor torque. For the no-load variant, the motor torque values fluctuated around zero, while for the variant with load, negative values during operation (ignoring the start-up phase) were assumed. The negative values here result from the assumed turning of generalized coordinates.

Application of the assumed control method requires the determination of certain model parameters (inertia, damping, elasticity) and the system's status (movement and speed).

The system settings selected by way of experiments had small errors. Due to the knowledge of the dynamics of the controlled object, in numerical studies the compensation function ensures very precise control; therefore, the control signal from the PD controller had much lower values. In real objects, with unknown disturbances, the level of compensation will not provide such perfect control and greater influence of the PD control should be expected. Moreover, the simulation did not consider signal processing delays that occur in real conditions, therefore, analysis of the results of empirical studies is necessary [49–53].

## 7. Experimental Research

Due to the use of a prototype test bench and the lack of appropriate certificates, empirical studies were performed on healthy people only. During tests, the trainees moved on a treadmill at a speed of 1.5 km/h. The set trajectory of the passive part was adjusted according to Equation (21). Before starting the device, an initial unloading of 100 N was applied. This preload force was obtained in manual control mode by controlling the drive system through the control panel. In the position obtained in this way, the coordinates of the motor shaft angle of rotation were reset. In the next step, the device was started and the trainee tried to move synchronously with the set trajectory of movement acting on the body weight support system with a variable load force. An increase in loading force was achieved by lowering the center of gravity of the trainee. The following parameters were recorded during the experimental tests:

- Set and realized positions of the passive and active parts of the body weight support system;
- Set and realized speeds of the passive and active parts of the body weight support system;
- Values of control signals;
- Follow-up movement errors;
- The value of the body weight support force (from the force sensor).

The parameters of the control system were the same as those obtained in the numerical research. The following figures present the results of the performed empirical studies.

The results of experimental research are shown in Figures 16–22. The adjusted and realized movements of the active part and passive parts in dynamic body weight support are shown in Figures 16 and 17. Figures 18 and 19 presents the adjusted and recorded movement speeds. Figure 20 presents the total control signal along with the PD controller signal and compensation control signal. Figure 21 specifies the follow-up movement errors. Figure 22 shows the value of body weight support force during the experiment as a function of time.

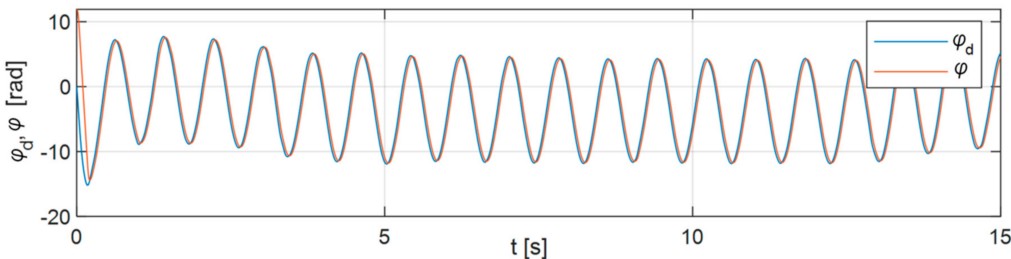

**Figure 16.** Set $\varphi_d$ and realized $\varphi$ positions of the active part as functions of time.

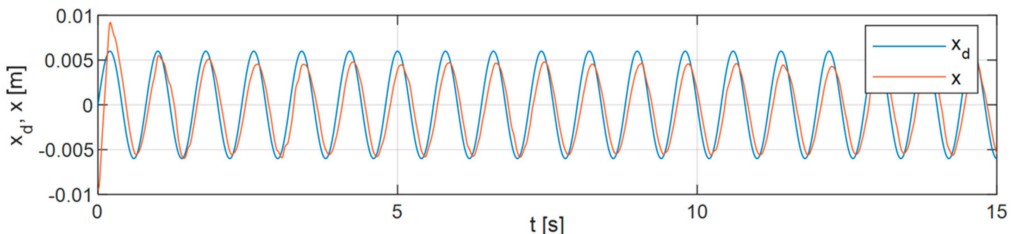

**Figure 17.** Set $x_d$ and realized $x$ positions of the passive part as functions of time.

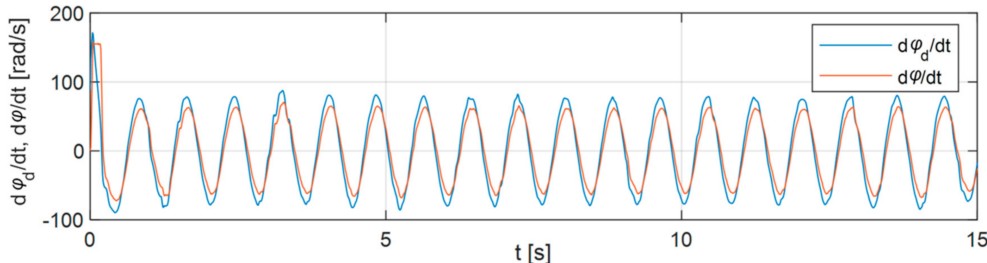

**Figure 18.** Set $d\varphi_d/dt$ and realized $d\varphi/dt$ angular speeds of the active part as functions of time.

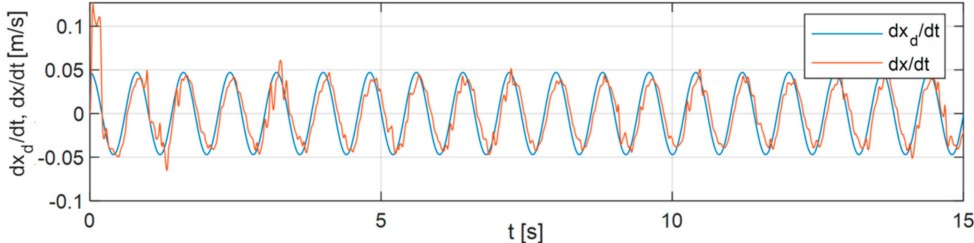

**Figure 19.** Set $dx_d/dt$ and realized $dx/dt$ speeds of the passive part as functions of time.

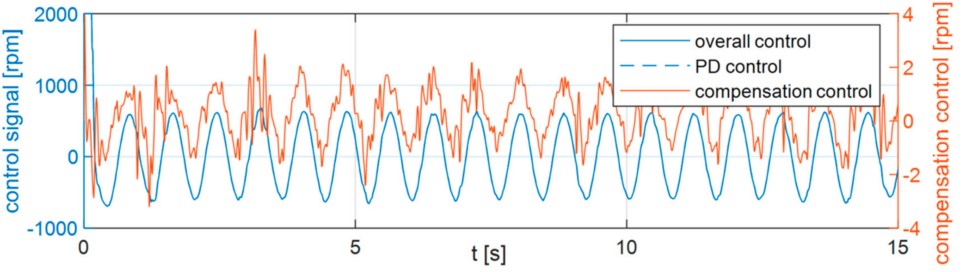

**Figure 20.** Signals of overall control and control signal components as functions of time.

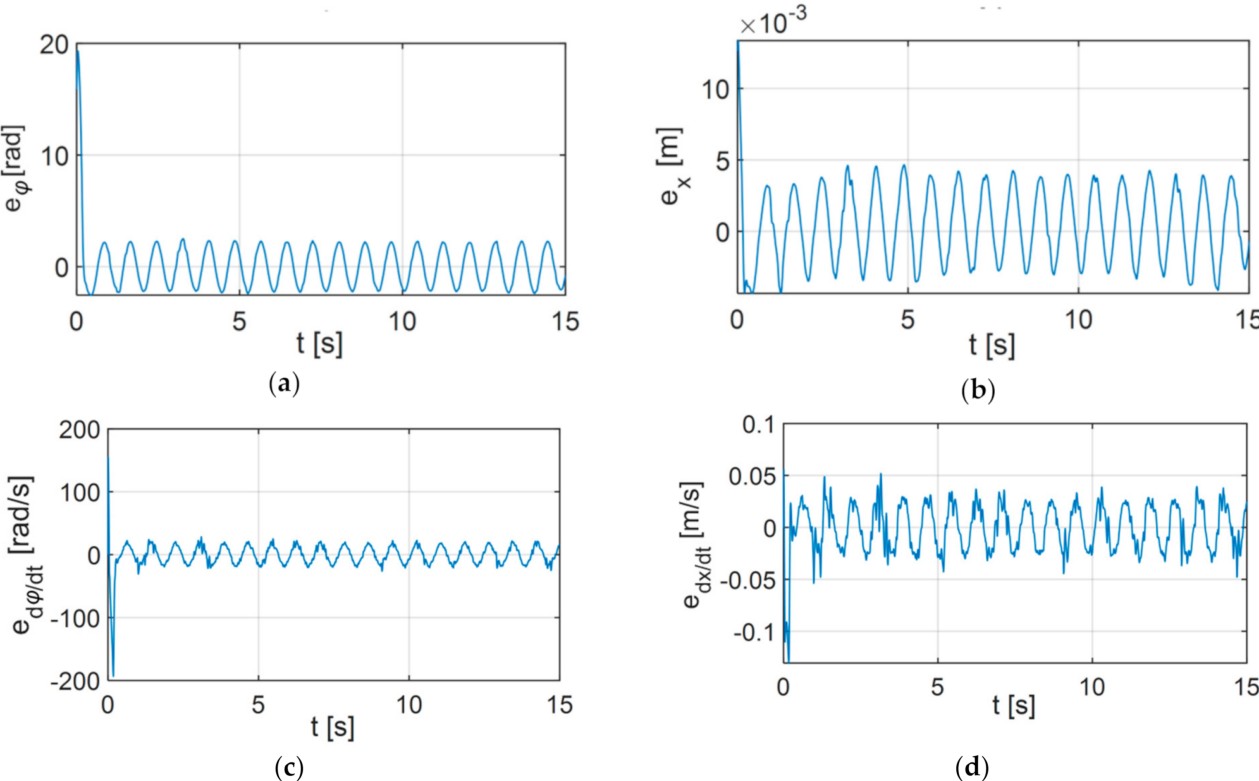

**Figure 21.** Follow-up movement errors: (**a**) position error of the active part $e_\varphi$; (**b**) position error of the passive part $e_x$; (**c**) speed error of the active part $e_{d\varphi/dt}$; (**d**) speed error of the passive part $e_{dx/dt}$.

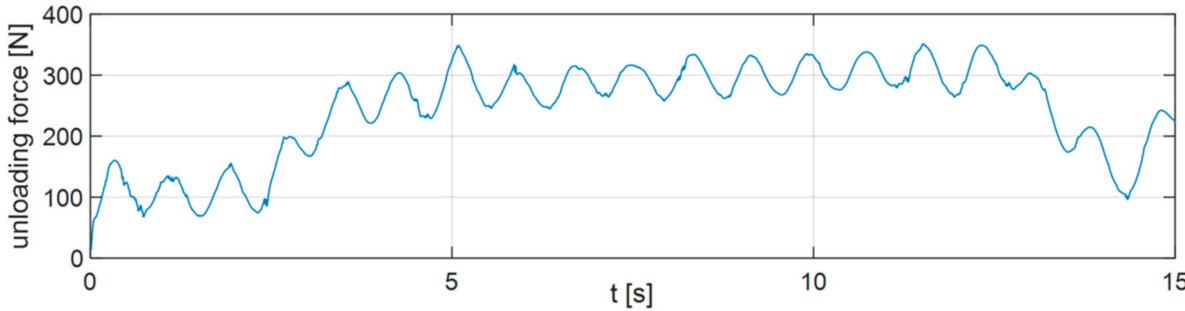

**Figure 22.** Value of the trainee's unloading force as a function of time.

The performed empirical studies confirmed the conclusions from the simulation tests related to the stable operation of the developed control system, regardless of disruptions caused by patient movements.

In the real object, servodrives are controlled in speed mode; therefore, the values of control signals cannot be directly compared with the results of numerical tests. Optimization of the regulator settings in a real facility will be possible after the implementation of a complete servodrive in the numerical model. The selection of settings during experimental tests is a time-consuming process; therefore, no further attempts were made to select them because the purpose of this study was only to demonstrate the stability of the proposed control system.

## 8. Summary and Conclusions

This paper includes the results of studies regarding the methodology used for tests on the stability of a control system for a dynamic body weight support system. It also describes the object of tests for which equations were formulated regarding a motion and control

method. The device has a control system based on a PD controller with compensation, which is essential considering its classification as an underactuated system. This work presents the complete algorithm for calculations used to describe the control system and to analyze the stability. The Lyapunov theory of stability function was also determined.

The presented methodology of calculations may be employed to analyze the stability of similar devices. However, knowledge of the mathematical model is required to determine the dynamic performance of a device and its parameters, as well as its control method.

The proposed control system features the trajectory of the active part, which is determined on the basis of movement of the passive part. During the performed tests, the trajectory of the passive part related to the adjustable function is formulated on the basis of previous studies of the gait kinematic parameters. It is also possible to modify the control system and use the measurement of the position of the passive part as the set trajectory. Alternatively, the force sensor, which is equivalent to an approach applied previously in similar devices, can be used [21,25,34]. In this case, the device performs a movement that follows the vertical movement of the patient only.

During the conducted tests, it was the body weight support system that forced patient gait thanks to the passive part movement. The use of such a control system in a rehabilitation device for the re-education of human gait is a new approach. Therefore, it is not possible to directly compare the obtained results with other works and the quality of control systems used so far in similar devices, e.g., with the control system of the ZeroG device, where only force controller occurs [21]. However, we know from the literature review that for underactuated mechanical systems, the use of the PD controller with compensation ensures better operation than the feedback loop with the PD controller itself; this has been shown inter alia on underactuated systems such as inverted pendulums, ball and beam systems, overhead cranes, or robots with flexible joints [54–60]. However, it is necessary to demonstrate the stability of such a control system so that the term associated with the compensation of the dynamics of the passive part does not cause destabilization [61–63]. The unstable control algorithm may result in dangerous movements of the drive in the body weight support system, which is particularly important due to the cooperation of the device with patients during rehabilitation. In the presented approach, the stability of the developed system has been demonstrated, which has not been proven so far in other studies on body weight support systems. However, in future studies it will be possible to compare the proposed algorithm with more advanced controllers, such as active disturbance rejection control, sliding mode control, or with adaptive algorithms.

On the basis of the results of empirical studies and the subjective opinion of the person performing the training, it is concluded that the use of this control system in a dynamic body weight support system forces the patient to adjust their walking cadence to the rope movement. Consequently, this control system has a significant impact on the person's gait kinematics. However, tests did not verify any real movements of the patient, which may be different in practice, e.g., due to the flexibility of the rope or harness. The usefulness of the proposed control method for rehabilitation purposes shall be determined by experts in the field of physiotherapy.

The proposed control system offers many possibilities for further modifications. Future work will focus on an extension to obtain a cascade whereby the control of the active part provides a compromise between the adjusted movement trajectory and the adjusted body weight support force. Similar force–position control solutions are widely employed in advanced mechatronic systems, e.g., in devices for upper and lower limb rehabilitation [64–67].

Another limitation of the conducted research was the performance of tests for the control system with experimentally selected settings. It should be noted that the operation of the device will be more accurate after the optimization process, for which the development of verified numerical models of servodrives are necessary.

**Author Contributions:** Conceptualization, G.G. and P.G.; methodology, G.G. and P.G.; software, P.G.; validation, G.G. and P.G.; formal analysis, S.D.; investigation, G.G., P.G. and S.D.; resources, S.D.; data curation, G.G. and P.G.; writing—original draft preparation, G.G. and P.G.; writing—review and editing, P.G.; visualization, G.G.; supervision, S.D.; project administration, S.D.; funding acquisition, S.D. All authors have read and agreed to the published version of the manuscript.

**Funding:** This work was financially supported by statutory funds from the Faculty of Mechanical Engineering of Silesian University of Technology in 2021.

**Institutional Review Board Statement:** Not applicable.

**Informed Consent Statement:** Informed consent was obtained from all subjects involved in the study.

**Data Availability Statement:** Data sharing not applicable.

**Conflicts of Interest:** The authors declare no conflict of interest.

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
