# Peer review of "Modeling and Control of an Underactuated System for Dynamic Body Weight Support"

_applsci, doi:10.3390/app11030905_

Round 1

Reviewer 1 Report

The manuscript was prepared without due care. It requires a thorough linguistic and editing correction, as well as improving the quality of figures (indication of elements in fig. 1, deletion of underlines in fig. 2).

The equations given as Eq.1 are given without any justification.

The problem addressed in the article is not new. There are many such devices, as well as the literature on the subject. For this reason, a "Discussion" section is absolutely required, where authors must indicate the reasons why their approach is new and what it is better than others - in other words, identify new elements.

Author Response

Please see the attachment. All changes in manuscript are marked with colored highlights.

Reviewer 2 Report

The authors focused on the stability analysis of the control system for the dynamic system based on the body weight support system powered by winches and Series Elastic Actuator (SEA) type drive. The results from the simulation and the experiment were also presented.

Here are some concerns regarding the methodology and results to improve the quality of the paper.

In section Control system modeling: The control law based on the PD controller with compensation represented the satisfaction of Lyapunov stability criteria. However, the control design actually based on sliding mode control. Why authors did not mention directly the sliding mode control system instead of the PD controller with compensation?

line 161: "where ... design constants" (consider revision)

line 173: "expression ... mean PD control method" (consider revision)

In section Simulation research: what are the differences between Figure 4 vs 8, Figure 5 vs 9, Figure 6 vs 10, and Figure 7 vs 11?

Furthermore, please provide the Figures with publication quality.

Line 301: "6. Experimental research" (consider revision)

In section Experimental research, please provide the experiment procedure in detail. Furthermore, the identified model parameters need to clarify. All control system parameters need to be provided.

Figure 21: please provide the Figures with publication quality.

Line 337: " t is also possible to" (consider revision)

Author Response

(The authors gave the same response as above.)

Round 2

Reviewer 1 Report

This is a sufficient revision. I recommend to accept the manuscript for publication.

Author Response

We would like to sincerely thank the Reviewer 1 for reviewing our article and for previous valuable suggestions.

We express our deep gratitude for the positive review of our article.

Reviewer 2 Report

The main content of this article focused on the stability analysis for the control system of the Body Weight Support system. However, a discussion regarding the advantages of the current control design and the comparison of the control quality with the other methods should be expressed.

Some minor revisions should be considered:

In Section: "6. Simulation research"
The subsection should be renumbered to be "6.1 Variant 1", ...
Please renumber Section "6. Experimental research " to be "7. Experimental research " since section number 6 existed already. In that case, the final section should be renumbered as well.
